

# Detection of hierarchical crowd activity structures in geographic point data

J. Miguel Salazar[1,*], Pablo López-Ramírez[1,*] and Oscar S. Siordia[2]

[1] Center for Research in Geospatial Information Sciences (Centrogeo), Tlalpan, Mexico City, Mexico
[2] National Geointeligence Laboratory, Merida, Yucatan, Mexico
[*] These authors contributed equally to this work.

## ABSTRACT

The pervasive adoption of GPS-enabled sensors has lead to an explosion on the amount of geolocated data that captures a wide range of social interactions. Part of this data can be conceptualized as event data, characterized by a single point signal at a given location and time. Event data has been used for several purposes such as anomaly detection and land use extraction, among others. To unlock the potential offered by the granularity of this new sources of data it is necessary to develop new analytical tools stemming from the intersection of computational science and geographical analysis. Our approach is to link the geographical concept of hierarchical scale structures with density based clustering in databases with noise to establish a common framework for the detection of crowd activity hierarchical structures in geographic point data. Our contribution is threefold: first, we develop a tool to generate synthetic data according to a distribution commonly found on geographic event data sets; second, we propose an improvement of the available methods for automatic parameter selection in density-based spatial clustering of applications with noise (DBSCAN) algorithm that allows its iterative application to uncover hierarchical scale structures on event databases and, lastly, we propose a framework for the evaluation of different algorithms to extract hierarchical scale structures. Our results show that our approach is successful both as a general framework for the comparison of crowd activity detection algorithms and, in the case of our automatic DBSCAN parameter selection algorithm, as a novel approach to uncover hierarchical structures in geographic point data sets.

# INTRODUCTION

Spatio-temporal analysis is a rapidly growing field within geographical information science (GIScience). The rate of increase in the amount of information gathered every day, the pervasiveness of Global Positioning System (GPS) enabled sensors, mobile phones, social networks and the Internet of Things (IoT), demand for robust and efficient analysis techniques that can help us find meaningful insights from large spatio-temporal databases. This diversity of digital footprints can be aggregated and analyzed to reveal significant emerging patterns (*Arribas-Bel, 2014*), but its accidental nature, produced as a side-effect of the daily operations of individuals, government agencies and corporations,

Corresponding authors
J. Miguel Salazar,
msalazar@centrogeo.edu.mx
Pablo López-Ramírez,
plopez@centrogeo.edu.mx

and not according to scientific criteria, calls for new analysis methods and theoretical approaches (*Zhu et al., 2017*; *Kitchin, 2013*; *Arribas-Bel, 2014*; *Liu et al., 2015*).

Within these new data sources, of special interest are those that can be characterized as spatio-temporal events data (*Kisilevich et al., 2010*), observations consisting of points in space and time and possibly with attribute information associated. Geo social media messages, cell phone calls, emergency services (911 reports), public service reports and criminal investigations are examples of this kind of data. Event data is linked to the social patterns of activity, they represent breadcrumbs that, when aggregated, can help us understand the underlying dynamics of the population. The production of this kind of event data is mediated by the activities people are undertaking and the geographic structure of space (*Jiang & Ren, 2019*).

The granularity of event data allows the researcher to generate arbitrary aggregations and analyze the data with different zoning schemes and scales (*Robertson & Feick, 2018*; *Zhu et al., 2017*). However, this freedom to build arbitrary aggregations comes at a cost, for example, the modifiable areal unit problem (MAUP) (*Openshaw, 1984*) links the results of analysis to the specific system of scales and zones used. Another closely related but different problem is the uncertain point observation problem (*Robertson & Feick, 2018*), which states the uncertainty of the assignation of a point observation to a given contextual area. As *Wolf et al. (2021)* points out, there have been important analytic developments to tackle the issues associated with MAUP, but this developments represent empirical answers to a problem that, as has been evident in the work of *Robertson & Feick (2018)* and *Kwan (2012)*, is in reality theoretically oriented: *solving* the MAUP through the development of optimal zoning schemes (*Bradley, Wikle & Holan, 2017*) does not automatically relate those zones to any geographically significant process or structure.

From a Computer Science perspective, the problem of aggregating individual observations into a system of zones has been tackled mainly through clustering algorithms (*Kisilevich et al., 2010*; *Frias-Martinez et al., 2012*; *Frias-Martinez & Frias-Martinez, 2014*; *Khan & Shahzamal, 2020*; *Liao, Yeh & Jeuken, 2019*; *Steiger, Resch & Zipf, 2016*). Clustering individual observations is, in the language of *Robertson & Feick (2018)*, an assignation of points into areal support and as such it implicitly involves a conceptualization of how the individual behaviors are structured to produce the patterns revealed by clusterization. In this sense, this data driven algorithms can be thought as belonging to the same empirical family of methods of optimal zoning algorithms, lacking theoretical support. As argued by *O'Sullivan (2017)* and *O'Sullivan & Manson (2015)*, this lack of grounding on formal geographic knowledge can often lead to spurious or irrelevant conclusions and, in general, hinder the advance of knowledge and the exploitation of new data sources for geographic analysis (*Arribas-Bel, 2014*).

One avenue of research, proposed in *Singleton & Arribas-Bel (2021)*, to tackle the problem of the use of data driven algorithms for the development of sound geographic analysis is to develop explicitly spacial algorithms that exploit our knowledge about the processes and structures that organize the spatial activity patterns of society. In our work, we tackle the problem of developing an algorithm that, through the theoretical concept of

hierarchical scales, is able to detect patterns that are geographically relevant and not only data driven.

Our contribution is hence threefold. First, since there is often no ground truth available to compare different clustering algorithms on geographical events data, we developed an algorithm to generate synthetic hierarchically structured data; second, we developed an algorithm for the automatic selection of the $\varepsilon$ parameter in DBSCAN that allows its use for the detection of density based hierarchical structures in geographic point data; and third, we propose a framework to compare the performance of different clustering algorithms. Our work sits at the intersection of computer science and geography, proving that approaching data driven problems from a theory oriented perspective, provides robust analysis frameworks.

The rest of the paper is organized as follows: In 'Detection of crowd activity scales and zones' we present the problem of detecting hierarchical crowd activity structures in geographic events databases; in 'Synthetic data', we describe the algorithm to generate synthetic samples to test different algorithms; in 'Clustering Algorithms' we describe the algorithms we are going to use; 'Experimental setting' describes the experimental setting used to compare different clustering algorithms; 'Evaluation' describes the metrics used to evaluate the performance of the clustering methods; in 'Results and discussion' we present our main results and finally in 'Conclusions and Further Work' we conclude and propose further research.

## DETECTION OF CROWD ACTIVITY SCALES AND ZONES

In the context of spatio-temporal events, as described by *Kisilevich et al. (2010)*, we are going to refer as crowd activity to the collective aggregated patterns observed in some spatio-temporal events data sets, specially in data describing some aspect of the behavior of human populations. Although not formally defined, this concept underpins most of the work that we are going to review in the rest of this section.

There is a substantial body of work on techniques for event detection using the geolocated Twitter feed (in this context, event refers to real world occurrences that unfold over space and time, which is different to the use of the term on spatio-temporal databases). *Atefeh & Khreich (2015)* present a survey of such techniques. Within this field, we are specially interested in works that use only the spatio-temporal signature of events and don't rely on attribute or content information (such as the content of Twitter messages), because this renders the methods easily translatable across different databases. Along this line, in *Lee, Wakamiya & Sumiya (2011)*, the authors propose a technique for detecting unusually crowded places by extracting the regular activity patterns performing K-means clustering over the geolocated tweets and then characterizing each cluster by the number of users, the amount of messages and a measure of the mobility of users in each cluster.

Another interesting research avenue is the detection of land use by accounting for the spatio temporal signature of events data sets. For example, in *Frias-Martinez & Frias-Martinez (2014)*, the authors develop a technique to extract, through self organized maps (*Kohonen, 1990*) and spectral clustering, different land uses in urban zones;

while *Lenormand et al. (2015)* use a functional, network based, approach to detect land use through cell phone records. Along the same line, *Lee, Wakamiya & Sumiya (2012)* develop a technique to extract significant crowd behavioural patterns and through them generate a characterization of the urban environment.

From this brief review we can infer some generalities involved in the extraction of crowd activity patterns from spatial spatio-temporal events data:

- Time is segmented in intervals and the definition of this intervals is arbitrary and not extracted from the data. In *Frias-Martinez & Frias-Martinez (2014)*, each day is divided in 20 min intervals; in *Lee, Wakamiya & Sumiya (2011)*, each day is segmented in four six hours intervals, while in *Lenormand et al. (2015)* each day is divided in hourly intervals.
- The geographic space is partitioned in a single scale tessellation of space.

Our work will focus on the second general characteristic: the way in which the space is partitioned to obtain crowd activity zones. In the reviewed works, the partitioning algorithm returns a single scale tessellation around the cluster centroids identified. This partition reflects the differences in point density across the whole space but, since it is flat (it has a single scale) it cannot represent the structures found at different scales, this means that such partitions mix the whole range of scales of the underlying data generating processes into a single tessellation.

However, when addressing crowd activity patterns from a geographic perspective, the issue of scale is evident: the underlying processes that generate the observed spatio-temporal distribution of events is organized as a hierarchy of scales, closely related to the urban fabric (*Jiang & Miao, 2015*; *Arcaute et al., 2016*; *Van Meeteren & Poorthuis, 2018*). This suggests the use of techniques for detecting crowd activity zones explicitly incorporating the concept of hierarchical scales. Along this line of work, in *Jiang & Ren (2019)* the authors prove that a hierarchical structure, based on the Natural Cities algorithm (*Jiang & Miao, 2015*), is able to predict the location of Twitter messages, while in *López-Ramírez, Molina-Villegas & Siordia (2019)*, the authors use a hierarchical structure to aggregate individual Twitter messages into *geographical documents* as input to latent topic models. This demonstrates the need to develop further methods to extract hierarchical crowd activity patterns from spatio-temporal events data.

Conceptually, hierarchical crowd activity detection algorithms are similar to hierarchical clustering algorithms. Crowd activity is characterized by arbitrary shaped agglomerations in space with potentially noisy samples, so the computational task is similar to the one tackled by DBSCAN (*Ester et al., 1996*). The main difference is that in the case of hierarchical crowd activity detection, one needs to find *structures within structures* to uncover the whole hierarchical structure. Notice that this problem is different from the problem of hierarchical clustering as approached by HDBSCAN (*McInnes, Healy & Astels, 2017*) or OPTICS (*Ankerst et al., 1999*), where the task is to find the most significative structures present in a data set, this will be further explained in 'Clustering Algorithms'.

A central problem to the development of algorithms to detect hierarchical crowd activity patterns is the lack of ground truth to test the results. For example, in *Arcaute et al. (2016)* and *Jiang & Ren (2019)*, the authors are interested in providing alternative,

tractable, definitions of cities and perform only qualitative comparisons with available data. In *López-Ramírez, Molina-Villegas & Siordia (2019)*, the authors extract regular activity patterns from the geolocated Twitter feed and also perform only a qualitative comparison to the known urban activity patterns. To provide an alternative that allows the quantitative evaluation of different algorithms to detect hierarchical crowd activity patterns, in the next section we describe an algorithm to generate synthetic data that aims to reproduce the most important characteristics found in real world event data.

## SYNTHETIC DATA

In order to test different hierarchical crowd activity detection algorithms, we developed a tool to generate synthetic data. The algorithm creates and populates a hierarchical cluster structure that reproduces the main characteristics of the structures we described in the previous sections. Our synthetic data generator creates a hierarchical structure by first creating a *cluster tree* where every node represents a cluster of points within a region delimited by a random polygon, within this polygon, a random number of clusters are generated, this procedure is carried on iteratively. For every level in the hierarchy we fill the space with noise points. Figure 1 shows an example of the tree structure generated by our algorithm as well as the polygons and points.

Geographic distributions that exhibit hierarchical structure have a characteristic heavy-tailed size distribution (*Jiang, 2013*; *Arcaute et al., 2016*; *Jiang & Ren, 2019*), having many more *small* objects than *large*. To show that our synthetic data exhibits this same property, we perform a Delaunay triangulation with the points as vertex, then obtain the lengths of the edges and sort them in descending order. We then proceed to select the length values larger than the mean (the Head) and the values smaller than the mean (the Tail), keeping only the latter in order to perform the Head-Tails break described in *Jiang (2013)*. Figure 2 shows three iterations of the Head-Tails break, each of them clearly exhibiting the heavy-tailed distribution characteristic of hierarchical scales. For each iteration we also calculate the HT-index (*Jiang & Yin, 2014*) and show that it corresponds with the level in the cluster tree as expected.

To facilitate research with our synthetic data generator, we developed a series of helper tools. For example, we include a tool to easily obtain the tags (to which cluster and level a point belongs) for every point and another tool to generate visualizations of the whole structure such as shown in Fig. 1. Our synthetic data can be used to test different clustering or aggregation algorithms in terms of their ability to detect the hierarchical structure in geographic point data sets. To allow the reproductibility of our research and make our methods available to other researchers, we published all our algorithms in a publicly available library in *Salazar, López-Ramrez & Sánchez-Siordia (2021)*.

Our clusters tree data structure can also be used to tag hierarchical clusters structures obtained by different algorithms. Instead of generating synthetic data, we can pass a hierarchical clusterization on a points sample to create a cluster tree by using the points and the clustering labels. Our helper tools simplify the process of comparison between different hierarchical crowd activity detection algorithms.

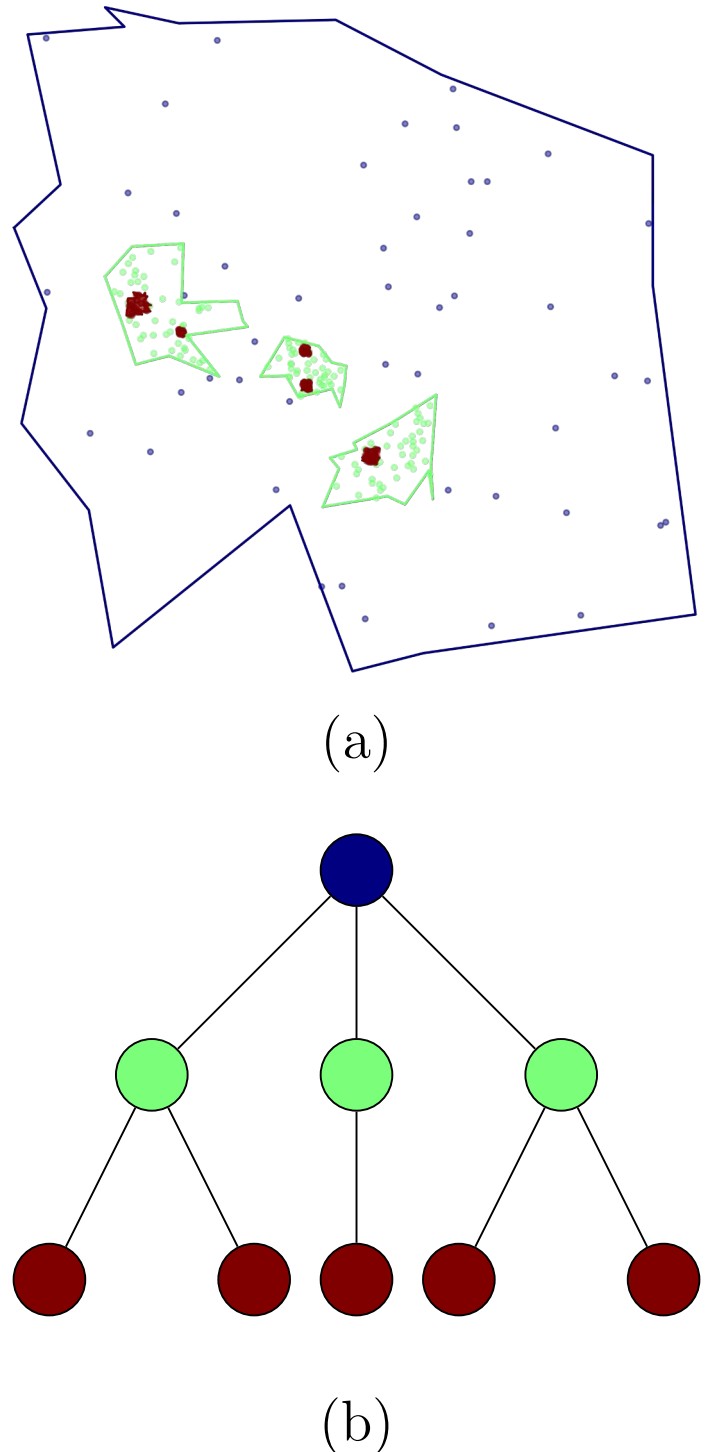

(a)

(b)

**Figure 1 Clusters obtained using the synthetic data generator (A) and the corresponding tree structure (B).** Nodes in (A) correspond with clusters in (B), the colors represent the different hierarchical levels.

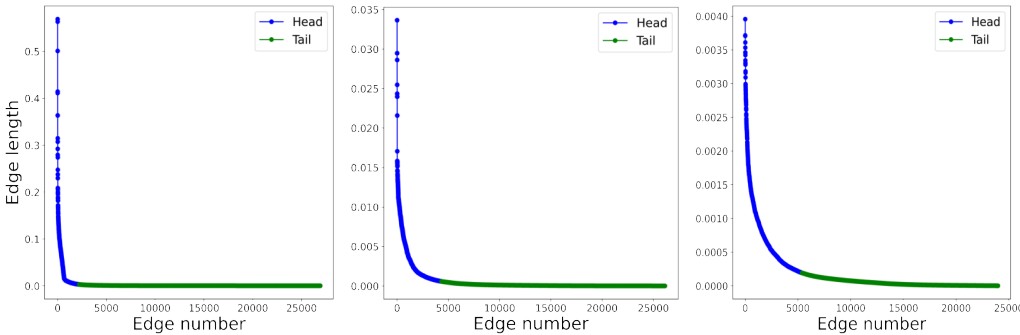

**Figure 2** **Length of the edges in the Delaunay triangulation for three successive levels in a synthetic data sample.** (A) First iteration of the Head-Tails break. The plot shows the length of the edges of the Delaunay triangulation obtained from the points. The edges are sorted according to the length value. (B) Second iteration of the Head-Tails break. The plot shows the length of the edges of the Delaunay triangulation obtained from the points. The edges are sorted according to the length value. (C) Third iteration of the Head-Tails break. The plot shows the length of the edges of the Delaunay triangulation obtained from the points. The edges are sorted according to the length value.

## CLUSTERING ALGORITHMS

There are several clustering algorithms available in the literature. For this research, based on the geographic considerations described in 'Detection of crowd activity scales and zones', we will focus on density based algorithms. This family of unsupervised learning algorithms identify distinctive clusters based on the idea that a cluster is a contiguous region with a high point density. Of special importance for our research is the ability of density based clustering algorithms to distinguish between cluster points and noise (*Ester et al., 1996*).

The result of a clustering algorithm strongly depends on parameter selection, thus, a common goal in the Computer Science literature has been the development of algorithms that use the least amount of free parameters, this has lead to the development of algorithms like OPTICS (*Ankerst et al., 1999*) and HDBSCAN (*Campello, Moulavi & Sander, 2013*; *McInnes, Healy & Astels, 2017*) that can detect hierarchical cluster structures with very little free parameters, this allows the automatic extraction of patterns from data without input from the researcher.

Hierarchical clustering algorithms focus on the task of detecting the most relevant density structures regardless of the scale. To illustrate this point, in Fig. 3 we show the original density clusters produced by our synthetic data generator together with the clusters obtained with HDBSCAN ('HDBSCAN') and our proposed Adaptative DBSCAN ('Adaptative DBSCAN'). As can be seen in the Figure, HDBSCAN detects the highest density clusters possible, this becomes even clearer by looking at the *Condensed Tree*, where the detected structures correspond to the deepest leafs in the tree: the most *persisting across scales* density structures. This focus on identifying persistent structures makes these algorithms unsuited for the task of detecting structures within structures because, as can be seen in the Figure, the intermediate scale levels are not detected since they do not persist in the *Condensed Tree*.

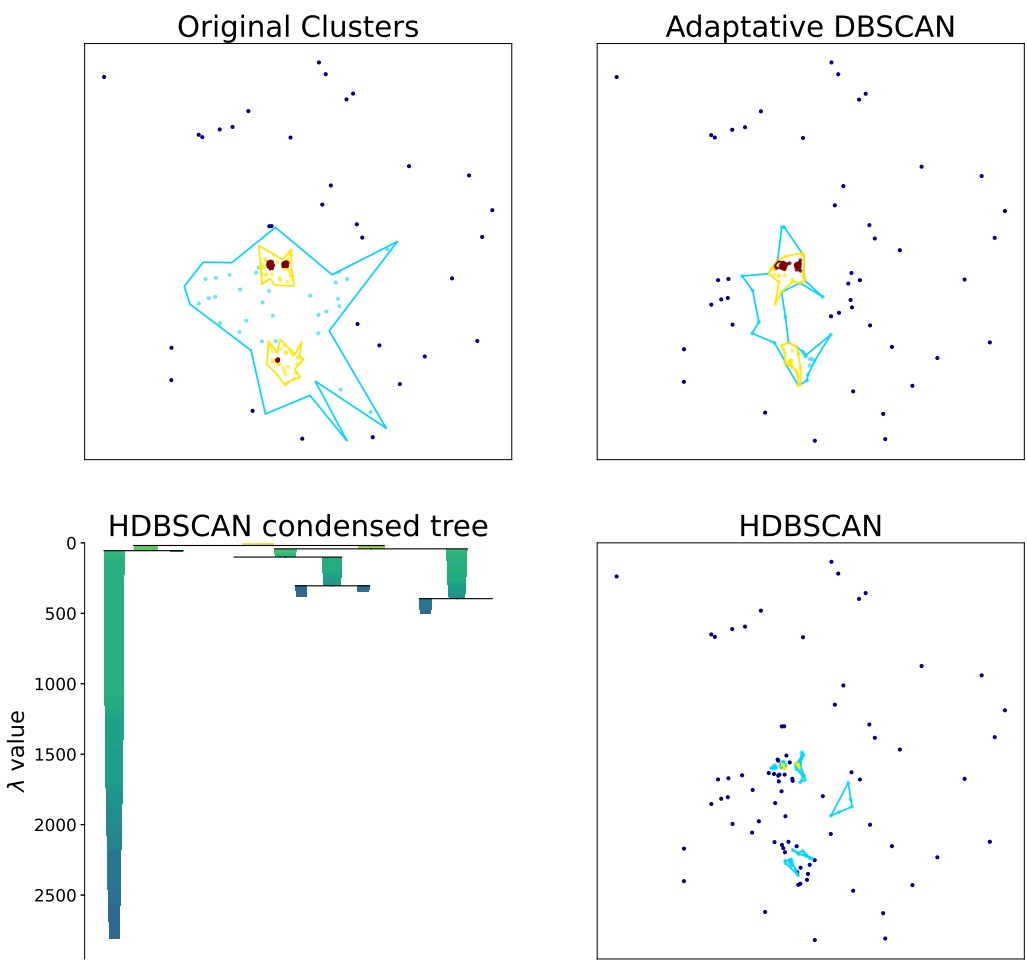

**Figure 3** **Cluster structure obtained with: synthetic data generator, Adaptative DBSCAN and HDB-SCAN. Bottom left figure shows the HDBSCAN condensed tree.** Original clusters produced with our synthetic data generator together with the clusters identified with HDBSCAN and Adaptative DBSCAN. The HDBSCAN Condensed Tree (bottom left) shows the depth of the structures detected, the two blue clusters shown in HDBSCAN results correspond to the leftmost and rightmost leafs.

Closely related to density based clustering, we have algorithms that extract hierarchical structures in point data by imposing thresholds to the distance between points. Hierarchical Percolation (*Arcaute et al., 2016*) and Natural Cities (*Jiang & Miao, 2015*) are examples of this kind of algorithms. The goal in this case is to explicitly extract the hierarchical structure implied by a heavy tailed distance distribution.

In the following Sections we present a brief description of the clustering and hierarchical scales extraction algorithms we are going to compare.

## Natural cities

The Natural Cities algorithm proposed in *Jiang & Miao (2015)* looks to objectively define and delineate human settlements or activities at different scales from large amounts of geographic information. The algorithm exploits the heavy tailed distribution of sizes

present in geographic data and uses Head-Tails breaks (*Jiang, 2013*) to iteratively extract the structures at successive scales. The algorithm can be summarized as follows:

- Use all the points and generate a Triangulated Irregular Network (TIN) using the Delaunay triangulation algorithm.
- Extract the edges of the triangulation and obtain their lengths.
- Calculate the average length.
- Use the average as a threshold to split the edges in two categories, the Head with those segments larger than the mean, and the Tail, with the segments smaller than the mean.
- Remove the Head.
- Generate the least amount of continuous polygons from the union of Tail edges.
- The points that are inside the obtained polygons are kept and the ones outside are consider noise.
- Repeat the procedure iteratively for each resulting polygon until the size distribution does not resemble a heavy tailed distribution.

At the end of the procedure, a hierarchical structure is obtained that contains structures within structures. This algorithms has been used to examine the spatial structure of road networks and their relation to social media messages (*Jiang & Ren, 2019*) and the evolution of Natural Cities from geo-tagged social media (*Jiang & Miao, 2015*).

## DBSCAN

DBSCAN is an algorithm designed to detect arbitrarily shaped density clusters in databases with noise (*Ester et al., 1996*). In the following we provide a brief description of the algorithm along with its core assumptions. The definitions established here will serve as basis to the description of OPTICS ('OPTICS'), HDBSCAN ('HDBSCAN') and Adaptive DBSCAN ('Adaptive DBSCAN').

DBSCAN uses a threshold distance (epsilon distance $\varepsilon$) and a minimum number of points *minPts* as initial parameters. From this, DBSCAN makes the following basic definitions:

**Definition** (Neighborhood). Let $p \in P$ be a point in the data set $P$ and $\varepsilon > 0$ a threshold distance, then the epsilon-neighborhood of $p$ is

$$\mathcal{N}_\varepsilon(p) = \{x | d(x, p) < \varepsilon\}$$

for $x \in P$

**Definition** (Reachable). A point $q \in P$ is a *reachable point* from a point $p \in P$ if there is a path of points $\{p_1, \ldots p_n - 1, q\} \subset P$ such that the distance between consecutive points is less than $\varepsilon$ $(d(p_i, p_{i+1}) < \varepsilon)$.

**Definition** (Core). Let $P$ be a set of points, a point $p$ is a *core* point of $P$ with respect to $\varepsilon$ and *minPts* if $|\mathcal{N}_\varepsilon(p)| \geq minPts$ where $\mathcal{N}_\varepsilon \subset P$.

**Definition** (Density reachable). A point $p$ is density reachable from a point $q$ with respect of $\varepsilon$ and *minPts* within a set of points $P$ if there is a path $\{p_1, \ldots p_n - 1, q\} \subset P$ such that $p_i \in P$ are core points of $P$ and $p_{i+1} \in N_\varepsilon(p_i)$.

**Definition** (Density connected). A point $p$ is *density connected* to a point $q$ with respect to $\varepsilon$ and *minPts* if there is a point $o$ such that both, $p$ and $q$ are density reachable from $o$ with respect to $\varepsilon$ and *minPts*.

**Definition** (Cluster). A cluster $C \subset P$ with respect to $\varepsilon$ and *minPts* is a non empty subset of $P$ that satisfies the following conditions:

1. $\forall p, q \in P$ : if $p \in C$ and $q$ is density connected from $p$ with respect to $\varepsilon$ and *minPts*, then $q \in C$.
2. $\forall p, q \in C$ $p$ is density connected to $q$ with respect to $\varepsilon$ and *minPts*.

**Definition** (Noise). Let $C_1, \ldots, C_k$ be the clusters of a set of points $P$ with respect to $\varepsilon$ and *minPts*. The *noise* is defined as the set of points in $P$ that do not belong to any cluster $C_i$.

Using the above definitions we can summarize the algorithm to cluster a set of points $P$ as:

- Consider the set $P_C = \{p_1, \ldots p_m\} \subset P$ of core points of $P$ with respect to $\varepsilon$ and *minPts*.
- For a point $p \in P_C$ take all the density connected points as part of the same cluster, remove these points from $P$ and $P_C$.
- Repeat the step above until $P_C$ is empty.
- If a point can not be reached from any core point is considered as *noise*.

In order to fit our aim of detecting structures within structures, we apply DBSCAN recursively to each of the discovered clusters. It is important to consider that the $\varepsilon$ parameter in DBSCAN is related to the relative point density of noise with respect to clusters, this means that in each recursive application one needs to set an appropriate value for $\varepsilon$; preserving the same value across iterations would not uncover density structures within clusters since those structures would, by definition, be of higher density than the encompassing cluster. In *López-Ramírez et al. (2018)*, $\varepsilon$ is decreased by a constant arbitrary factor on each iteration, in 'Adaptive DBSCAN' we propose a novel method for automatically selecting $\varepsilon$ for each cluster and iteration that overcomes this limitation.

## OPTICS

The OPTICS algorithm is fully described in *Ankerst et al. (1999)*, here we will make a brief presentation of its main characteristics, adapted to our research problem. OPTICS works in a similar fashion to DBSCAN and can be seen as an extension of it. The algorithm uses the notion of density-based cluster ordering to extract the corresponding density-based cluster for each point. OPTICS extends the definitions used in DBSCAN by adding:

**Definition** (Core distance). The *core distance* $(d_{core}(p, \varepsilon, minPts))$ of a point $p$ with respect to $\varepsilon$ and *minPts*, is the smallest distance $\varepsilon' < \varepsilon$ such that $p$ is considered a core point with respect to *minPts* and $\varepsilon'$. If $|\mathcal{N}_\varepsilon(p)| < minPts$ then the core distance is undefined.

**Definition** (Reachability distance). Let $p$ and $o$ be points in a set $P$ and take $\mathcal{N}_\varepsilon(o)$. The *reachability-distance* $(reach - dist_{(\varepsilon, minpts)}(p, o))$ is undefined if $|\mathcal{N}_\varepsilon(o)| < minPts$, else is $max(d_{core}(p, \varepsilon, minPts), d(p, o))$.

Let $P$ be a set of points, and set values for $\varepsilon$ and *minPts*. The OPTICS algorithm can be summarized as following:

- For each point $p \in P$ set the reachability-distance value as UNDEFINED ($L_{reach-dis}(p) =$ UNDEFINED $\forall p \in D$).
- Set a empty processed list $L_{Pro}$.
- Set a empty priority queue $S$
- For each unprocessed point $p$ in $P$ do:

  – Obtain $\mathcal{N}_\varepsilon(p)$
  – Mark $p$ as processed (add $p$ to $L_{Pro}$)
  – Push $p$ priority queue $S$
  – If $d_{core}(p, \varepsilon, minPts)$ is UNDEFINED move to the next unprocessed point. If not then:

    * Update the queue $S$ based on the rechability-distance by the *update* function and use $\mathcal{N}_\varepsilon(p)$, $p$, $\varepsilon$, and $minPts$ as input parameters.
    * For each $q$ in $S$ do:

      · Obtain $\mathcal{N}_\varepsilon(q)$
      · Mark $q$ as processed (add $q$ to $L_{Pro}$).
      · Push $q$ in the priority queue $S$.
      · If the $d_{core}(p, \varepsilon, minPts)$ is not UNDEFINED, use the *update* function with the $\mathcal{N}_\varepsilon(q)$, $q$, $\varepsilon$, and $minPts$ as parameters to update $S$.

  – The algorithm expands $S$ until no points can be added.

The *update* function for a priority queue $S$, uses as parameters a neighborhood $\mathcal{N}_\varepsilon(p)$, the center of the neighborhood $p$, and the $\varepsilon$ and $minPts$ values, and is defined as:

- Get the core-distance of $p$ ($d_{core}(p, \varepsilon, minPts)$).
- For all $o \in \mathcal{N}_\varepsilon(p)$ if $o$ is not processed, then

  – Define a new reachability-distance as $new_{reach-dis} = max(d_{core}(p, \varepsilon, minPts), d(p, o))$
  – If $L_{reach-dis}(o)$ is UNDEFINED (is not in S) then $L_{reach-dis}(o) = new_{reach-dis}$ and insert $o$ in $S$ with value $new_{reach-dis}$.
  – If $L_{reach-dis}(o)$ is not UNDEFINED ($o$ is already in S), then if $new_{reach-dis} < L_{reach-dis}(o)$ update the position of $o$ in $S$ moving forward with the new value $new_{reach-dis}$.

After this, we have reachability-distance values ($L_{reach-dis} : D \rightarrow \mathbb{R} \cup$ UNDEFINED) for every point in $P$ and an ordered queue $S$ with respect to $\varepsilon$ and $minPts$. Using this queue, the clusters are extracted using $\varepsilon' < \varepsilon$ distance and $minPts$ by assigning the cluster membership depending on the reachability-distance.

To decide whether a given point is noise or the first computed element in a cluster, is necessary to define that UNDEFINED $> \varepsilon > \varepsilon'$. After this, the assignation of the elements of $S$ is performed as:

- Set the $Cluster_{ID} =$ NOISE.
- Consider the points $p \in S$.
- If $L_{reach-dis}(p) > \varepsilon'$ then,

  – If $d_{core}(p, \varepsilon', minPts) \leq \varepsilon'$, the value of $Cluster_{ID}$ is set to the next value, and the cluster for the element $p$ is set as $Cluster_{ID}$.

- If $d_{core}(p, \varepsilon', minPts) \geq \varepsilon'$, the point $p$ is consider as NOISE.

- If $L_{reach-dis}(p) \leq \varepsilon'$, the cluster for the element $p$ is $Cluster_{ID}$.

The order of $S$ guarantees that all the elements in the same cluster are close. The $L_{reach-dis}$ values allow us to distinguish between clusters, these clusters will depend on the $\varepsilon'$ parameter.

In our work we use a recursive application of OPTICS to uncover the hierarchical scale structure, in the same fashion as *López-Ramírez et al. (2018)*.

## HDBSCAN

The algorithm *Density Based Clustering based on Hierarchical Density Estimates* is presented in *Campello, Moulavi & Sander (2013)*. The main intuition behind HDBSCAN is that the most significant clusters are those that are preserved along the hierarchical distribution. Before developing an explanation of HDBSCAN we need the following definitions:

**Definition** (Core k-distance). Let $p$ be a point in space, the core distance $d_{core_k}(p)$ of $k$ is defined by $d_{core_k}(p) = \max(\{d(x, p) | x \in \text{k-NN}(p) \text{ if } x \neq p\})$ where k-NN$(p)$ is the set of the $k$-nearest neighbors of $p$ for a specific $k \in \mathbb{N}$.

**Definition** (Core k-distance point). A point $p$ is a Core $k$-distance point with respect to $\varepsilon$ and $minPts$, if the number of elements in the core neighborhood of $p$ ($\mathcal{N}_{core_k}(p, \varepsilon, minPts) = \{x | d_{core}(x, p) < \varepsilon\}$) is greater or equal than $minPts$. Where $\varepsilon$ is greater or equal to the core distance of $p$ for a given $k$ value.

**Definition** (Mutual Reachability Distance). For every $p$ and $q$ the Mutual Reachability Distance is $d_{mreach}(p, q) = \max(\{d_{core_k}(p), d_{core_k}(q), d(p, q)\})$

Consider the weighted graph with every point as vertices and the mutual reachability distance as weights. The resulting graph will be strongly connected, the idea is to find islands within the graph that will be consider clusters.

To classify the vertices of the graph in different islands its easier to determine which do not belong to the same island. This is done by removing edges with larger weights than a threshold value $\varepsilon$, by reducing the threshold the graph will start to disconnect and the connected components will be the islands (clusters) to consider. If a vertex in the graph is disconnected from the graph or the connected component doesn't have the $minPts$, the corresponding vertex are considered noise. The result will depend on $\varepsilon$.

To reduce computing time *HDBSCAN* finds a Minimum spanning tree (MST) of the complete graph. The tree is built one edge at a time by adding the edge with the lowest weight that connects the current tree to a vertex not yet in the tree.

Using the MST and adding a self edge to all the vertices in the graph with a distance core value($MST_{ext}$), a dendrogram with a hierarchy is extracted. The HDBSCAN hierarchy is extracted using the following rules (*Campello, Moulavi & Sander, 2013*):

- For the root of the tree assign all objects the same label (single cluster)
- Iteratively remove all edges from $MST_{ext}$ in decreasing order of weights (in case of ties, edges must be removed simultaneously):

  - Before each removal, set the dendrogram scale value of the current hierarchical level as the weight of the edge(s) to be removed.

– After each removal, assign labels to the connected component(s) that contain(s) the end vertex(-ices) of the removed edge(s), to obtain the next hierarchical level: assign a new cluster label to a component if it still has at least one edge, else assign it a null label (noise).

The clusters thus obtained from the dendrogram depend on the selection of a $\lambda$ parameter based on the estimation of the stability of a cluster. This is done using a probability density function approximated by the $k$-nearest neighbors.

Once again, instead of relying on the hierarchical structure obtained from HDBSCAN, to discover our structures within structures we apply the algorithm recursively to every cluster.

## Adaptative DBSCAN

In *López-Ramírez et al. (2018)* the authors propose the recursive application of DBSCAN as a suitable algorithm to uncover the hierarchical structure present in spatio-temporal events data sets. The main drawbacks of this proposal are, on the one hand, the need to select appropriate $\varepsilon$ and *minPts* values for each recursive application of DBSCAN and, on the other hand, that once this values are selected they are used for every cluster in a given hierarchical level, thus assuming that every cluster has the same intrinsic density properties. In this paper we propose an algorithm to automatically select $\varepsilon$ values for each cluster, this algorithm draws on methods proposed in the available literature and adapts them to the problem of identifying clusters on hierarchically structured geographical data.

Before explaining our algorithm, let us review how *minPts* and $\varepsilon$ values are estimated. For the *minPts* parameter, the general rule is to select it so $minPts \geq D+1$, where $D$ is the dimension of the data, so in the recursive application of DBSCAN this parameter is fixed. In general, the problem of selecting $\varepsilon$ is solved heuristically by the following procedure:

- Select appropriate *minPts*.
- For each point in the data get the $k$th-nearest neighbors using *minPts* as the value of $K$.
- The different distances obtained are sorted smallest to largest (Fig. 4, *K-Sorted* distance graph).
- Good values of $\varepsilon$ distance are those where there is a big increment in the distance. This increment will correspond to an increment on the curvature in the plot, this point is called the elbow.

An automatic procedure to select $\varepsilon$ is presented in *Starczewski, Goetzen & Er (2020)*, where the authors propose a mathematical formulation to obtain the elbow value of the *K-Sorted* distance graph once *minPts* is given. This formulation tends to find the highest density clusters present in the data, so when applied recursively to hierarchically structured data it will tend to find the clusters in the deepest hierarchical level and will label as noise points belonging to intermediate hierarchical levels.

To overcome this limitation, we propose an alternative procedure to automatically select $\varepsilon$ that leads to larger distance values and thus less dense clusters in each recursive application of DBSCAN. First, notice that either on the heuristic described above or in the procedure by *Starczewski, Goetzen & Er (2020)*, the $\varepsilon$ value depends on the selection

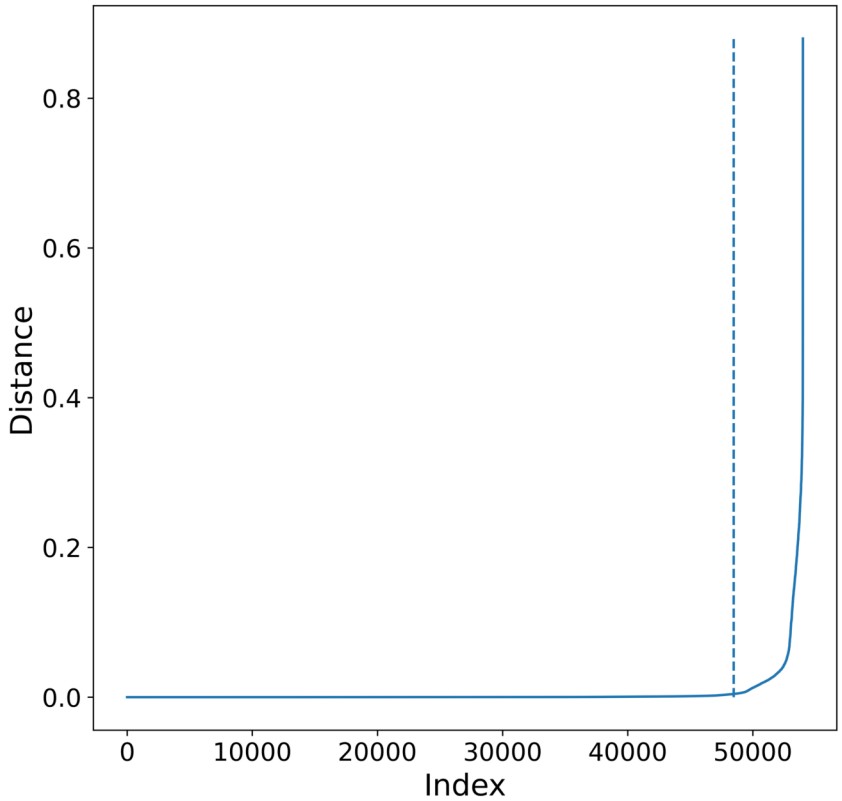

**Figure 4  K-sorted distance graph for $K = 5$.** K-sorted distance graph (for $k = 5$) showing with a dotted line the location of the elbow value obtained by the procedure outlined in *Satopaa et al. (2011)*.

of $K$, larger values of $K$ will in general lead to larger $\varepsilon$ values, although not in a strictly monotonic way as seen in Fig. 5. In principle it is possible to select $K$ as the maximum number of points in the data set and obtain a suitable $\varepsilon$ value, the problem is that for large data sets this is not feasible due to computational constraints.

To reduce the computational cost of finding appropriate $\varepsilon$ values, we observe in Fig. 6 the similarities in the *K-Sorted* distance graph for a range of $K$ values, it can be seen that the elbow value is similar for close values of $K$. Using this observation we obtain the *K-Sorted* distance graph only for a sample $K_s \subset \{1, \ldots, N\}$, where $N$ is the number of points, instead of every possible value, thus reducing the time and computational cost needed to obtain a suitable $\varepsilon$ distance value.

To find the elbow value for a given $k \in K_s$, we use the library developed in *Satopaa et al. (2011)*. The algorithm is $\mathcal{O}(n^2)$ so reducing the points involved in this step is also important. For each $k$, the algorithm takes as input the $N * k$ distances. To limit the number of distances passed to the algorithm, we average the $N * k$ distances in bins of size $k$, thus we only use $N$ values to calculate the elbow.

Our adaptative DBSCAN algorithm can be summarized as follows:

- Let $N$ be the number of points in the data set.

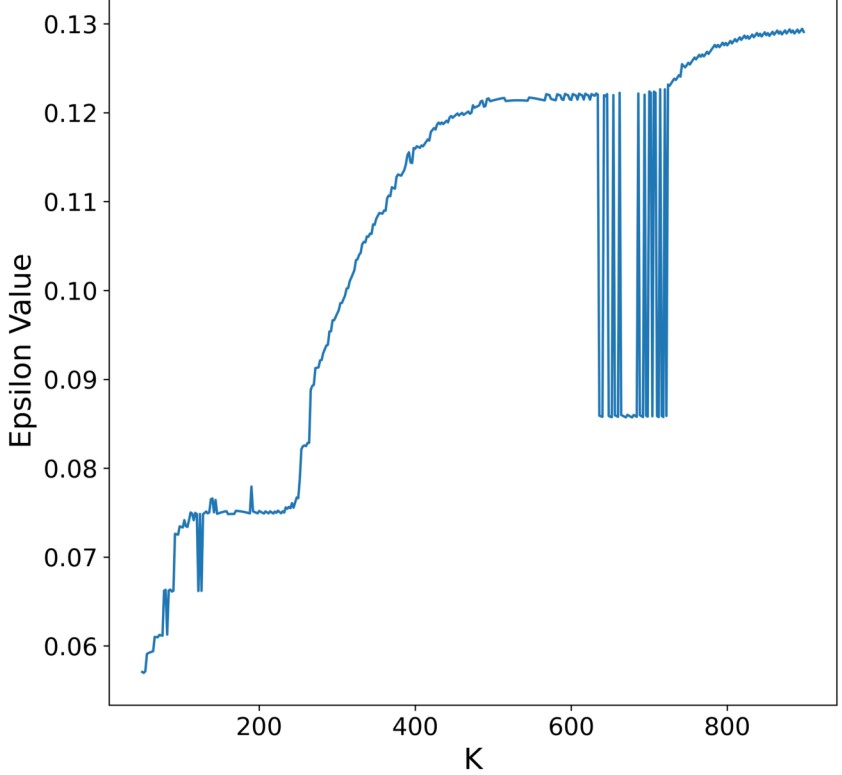

**Figure 5  Relationship between ε and the number of neighbors.** ε values obtained by applying the procedure outlined in *Satopaa et al. (2011)* to a synthetic data set for every possible value of $K$.

- For every step $k$ in the range $\{minPts, \ldots, \lfloor \frac{N}{10} \rfloor\}$ obtain the following:
    - For each point $p$, obtain the distances to the $k$ nearest neighbors.
    - Sort the distances.
    - Divide the distance interval in bins of size $k$ and obtain the average for each bin.
    - Get the elbow value for all the average bins as in *Satopaa et al. (2011)*.
- Take $\varepsilon$ as the maximum of all elbow values from the previous step.

This algorithm for calculating a suitable $\varepsilon$ value for each identified cluster allow us to find clusters with intermediate densities (correspondingly to intermediate hierarchical levels) and, at the same time, do it in a computationally efficient manner.

## EXPERIMENTAL SETTING

In order to develop a common framework for the comparison of the different clustering algorithms considered, we use the synthetic data generation algorithm described in 'Synthetic data' to generate random scenarios with 3, 4 and 5 levels, for each number of levels we carried out at least 100 experiments. By definition, each random scenario has a different number of clusters per level and thus different numbers of children per cluster, this provides the algorithms with a wide range of situations to tackle. Each one of the

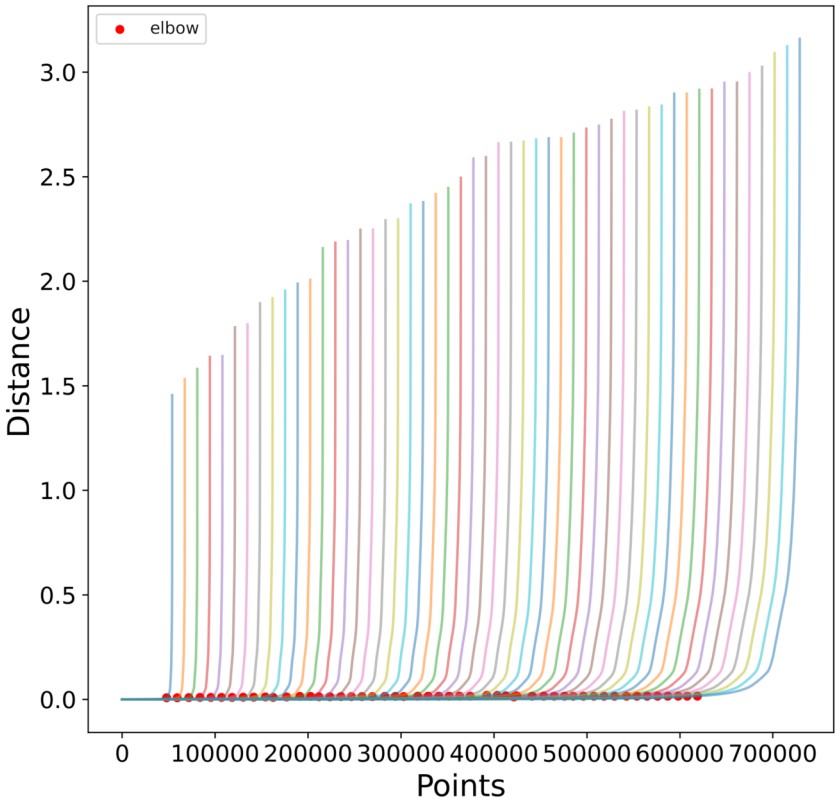

**Figure 6** **K-Sorted distance graphs for a range of K values.** The red dots show the elbow value calculated as in *Satopaa et al. (2011)*. Notice how close values of K, produce similar $\varepsilon$ values.

scenarios is passed trough the algorithms described in 'Clustering Algorithms' to generate statistically significant evaluations of each algorithm along every metric. The algorithms are evaluated using the metrics described in 'Evaluation'.

For the DBSCAN and OPTICS algorithms, the implementations used are from *Pedregosa et al. (2011)*. The HDBSCAN used is based on *McInnes, Healy & Astels (2017)*, and the Natural cities and adaptive-DBSCAN are our own implementations.

For all the algorithms the stopping conditions (in the recursive application) are the same: when the number of points in a cluster falls below a fixed *minPoints*, no further iterations are performed for said cluster.

## EVALUATION

We will evaluate the performance of the clustering algorithms along three complementary tasks: first, as global (across levels) classification algorithms, evaluating the ability of each algorithm to distinguish globally between cluster and noise points; second as per level classification algorithms, evaluating the ability to distinguish, for each level, noise and cluster points; finally, we will introduce a metric to compare the resulting shapes of the clusters obtained at each level. These three tasks are all important in the detection of hierarchical crowd activity structures: the first inform us about the global distinction

between activity and noise; the second does a similar job but at each level, while the last is very important from a geographical pattern point of view since it inform us about the similarity in the shapes of the activity structures detected.

## Global evaluation

To evaluate the performance of the different clustering algorithms as global classification algorithms, we label the obtained clusters using our tree cluster structure. In Fig. 7 we show an example output for the different algorithms. For every level we label points as either belonging to a cluster or as noise. This labeling has the property that when a point is labeled as noise in level $n$, then for every level $n+j$ with $j \geq 1$ the point will be tagged as noise. Thus we can obtain a label for every point as the concatenation of $NOISE, SIGNAL$ tags. For example, in a three level cluster tree, the possible labels are the following: $SIGNAL\_SIGNAL\_SIGNAL$, $SIGNAL\_SIGNAL\_NOISE$, $SIGNAL\_NOISE\_NOISE$, $NOISE\_NOISE\_NOISE$.

To evaluate the algorithms, we will use the Normalized Mutual Information ($NMI$) between the algorithm classification and the ground truth. $NMI$ has the advantage of being independent of the number of classes (*Vinh, Epps & Bailey, 2009*), thus giving a fair evaluation of the performance of the algorithms across experiments.

**Definition** (Normalized Mutual Information). $NMI$ is defined as:

$$NMI(X,Y) = \frac{2I(X,Y)}{H(X)+H(Y)}$$

where $P(x)$ is the probability to get the label $x$, $H(X)$ is the Shannon entropy of the set of labels $X$ and $I(X,Y)$ is the mutual information between the sets of labels $X$ an $Y$.

**Definition** (Shannon entropy). The Shannon entropy of a set of labels $X$ is defined as:

$$H(X) = -\sum_{x \in X} P(x) \log P(x)$$

where $P(x)$ is the probability of label $x$.

**Definition** (Mutual information). Let $X = \{X_1, \ldots, X_l\}$ and $Y = \{Y_1, \ldots Y_n\}$ be two sets of labels for the same set of points ($N$), the mutual information between the two set of labels is calculated as:

$$I(X,Y) = \sum_{i=1}^{|X|} \sum_{j=1}^{|Y|} P(i,j) * log\left(\frac{P(i,j)}{P(i)*P'(j)}\right)$$

where $P(i) = |X_i|/|N|$ is the probability of a random point belonging to class $X_i$, $P'(j) = |Y_i|/|N|$. And $P(i,j) = \frac{|X_i \cap Y_j|}{|N|}$ is the probability that a random point belongs to both classes $X_i$ and $Y_j$.

## Per level evaluation

To reflect the iterative clustering process, we evaluate the classification obtained for each level. In order to carry out this evaluation, we label, for each level, the points as either $SIGNAL$ for points belonging to a cluster or $NOISE$ otherwise. For each level $n$ we include only the points that have the $SIGNAL$ label in $n-1$ level since those are the only points seen

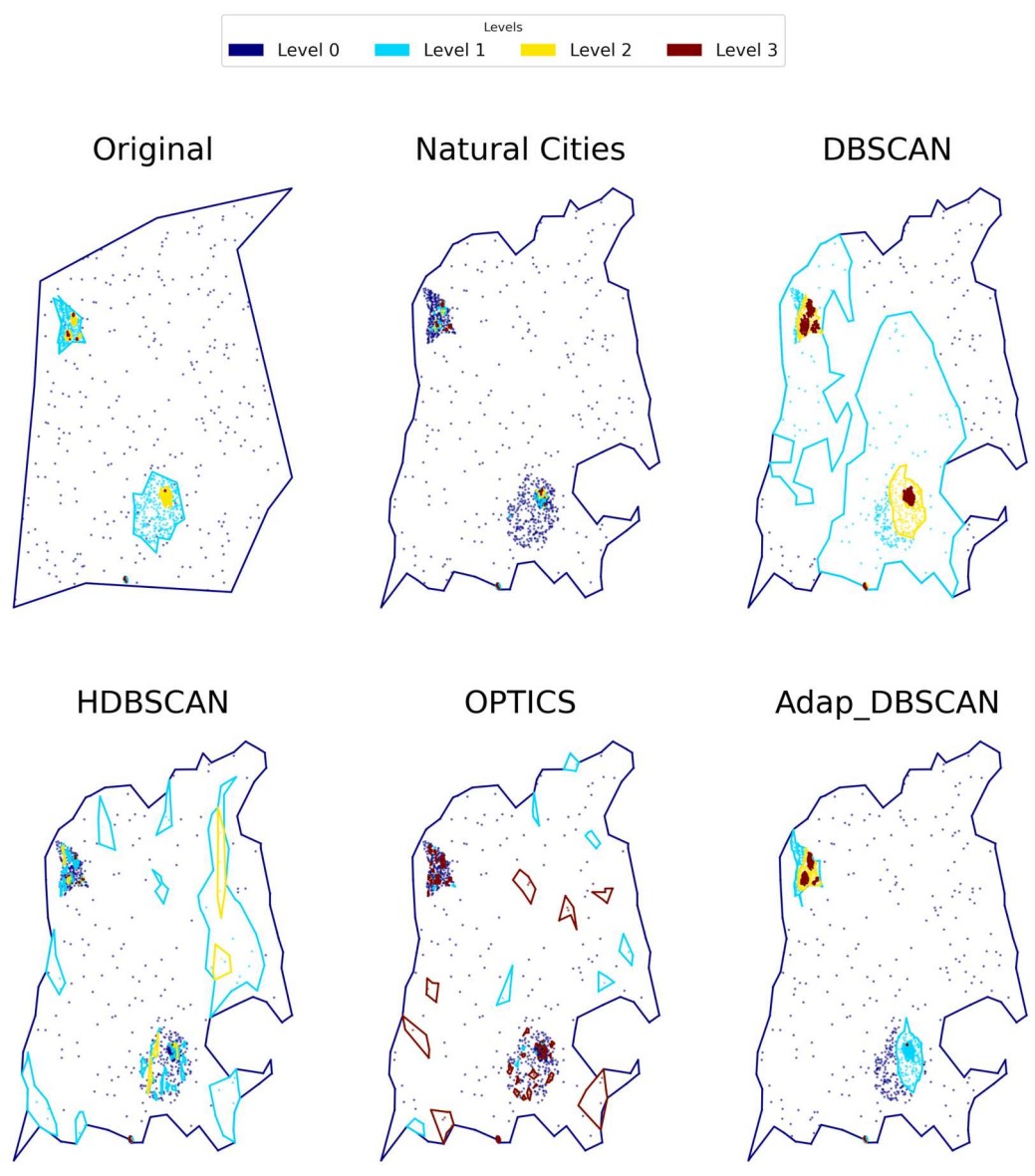

**Figure 7** **Results of different hierarchical clustering algorithms on a synthetic data set.** The clusters obtained with the different algorithms are represented by colored polygons (each hierarchical level is represented by a single color), the points shown the corresponding noise in each level.

by the algorithm at each recursive iteration. Using this approach, the per level evaluation corresponds to the clusters obtained on level *n* and is performed only for the points that correspond to each level.

In general, the labels obtained for each level are unbalanced with more samples of the noise class. Therefore for the evaluation we use the Balanced Accuracy (*BA*) for binary classification (*Brodersen et al., 2010*) on each level.

**Definition** (Balanced accuracy). Let $X$ be the ground truth labels in a point set $N$ and $Y$ the set of predicted labels in the same set. The balanced accuracy for a binary labeling is defined as:

$$BA(X,Y) = \frac{1}{2}\left(\frac{TP}{TP+FN} + \frac{TN}{TN+FP}\right)$$

where $TP$ is the true positive set of labels, $TN$ the true negative set of labels, $FN$ is the false negative set of labels, and $FP$ is the false positive set of labels.

## Shape evaluation

For geographic applications such as those described in 'Detection of crowd activity scales and zones', it is important to also evaluate the shape of the clusters obtained. Therefore we propose a measure that compares the shapes of the clusters obtained by each algorithm.

Our Similarity Shape Measure (SSM) compares the shapes of the Concave Hulls of each cluster, obtained by the optimal alpha-shape of the point set (*Edelsbrunner, 1992*; *Bernardini & Bajaj, 1997*). Thus, for our *SSM*, each cluster is represented by a polygon.

A simple way to compare polygon shapes is the Jaccard Index, mostly used in computer vision to compare detection algorithms.

**Definition.** (Jaccard Index) The Jaccard Index or Jaccard Similarity Coefficient between polygons $P$ and $Q$ is defined as:

$$Jacc(P,Q) = \frac{Area(P \cap Q)}{Area(P \cup Q)}$$

where *Area* is the area of the polygon.

The main issue in implementing a shape similarity measure is in determining which polygons to compare in each level. As can be seen in Fig. 7, every algorithm outputs different number and shapes of polygons in each level so there is no straightforward way of assigning corresponding polygons across algorithms (or the ground truth). To overcome this issue, our Similarity Shape Measure compares, using the Jaccard Index, the polygons produced by an algorithm with the polygons in the ground truth, weighting the index by the number of points in each polygon and its corresponding intersections.

**Definition** (Similarity Shape Measurement). Let $\mathcal{P}_i^O = \{P_0^O, \dots P_n^O\}$ be the set of polygons for the clusters on the $i-th$ level in the ground truth and $\mathcal{Q}_i^C = \{Q_0^C, \dots Q_m^C\}$ the polygons to evaluate for the same level, then the similarity between $\mathcal{P}_i^O$ and $\mathcal{Q}_i^C$ is:

$$SSM(\mathcal{P}_i^O, \mathcal{Q}_i^C) = \frac{\sum_{P_l^O \in \mathcal{P}_i^O Q_k^C \in Q_i^C} |P_l^O \cap Q_k^C| * Jacc(P_l^O, Q_k^C)}{\sum_{P \in \mathcal{P}_i^O} |P| + \sum_{Q \in \mathcal{Q}_{not}} |Q|}$$

where $\mathcal{Q}_{not} = \{Q | Q \cap P_i^O = \emptyset\} \subset \mathcal{Q}_i^C$ for all $P_i^O \in \mathcal{P}_i^O$.

The *SSM* takes values in $[0,1]$, the maximum value corresponds to the case when all polygons that intersect across the evaluation sample and the ground truth satisfy $Jacc(P_l^O, Q_k^C) = 1$, all the points within the polygons $P_l^O$ are in a corresponding polygon $Q_k^C$ and the family $\mathcal{Q}_{not}$ is empty. Conversely, *SSM* takes the value 0 when no polygons intersect across the evaluated algorithm and the ground truth.

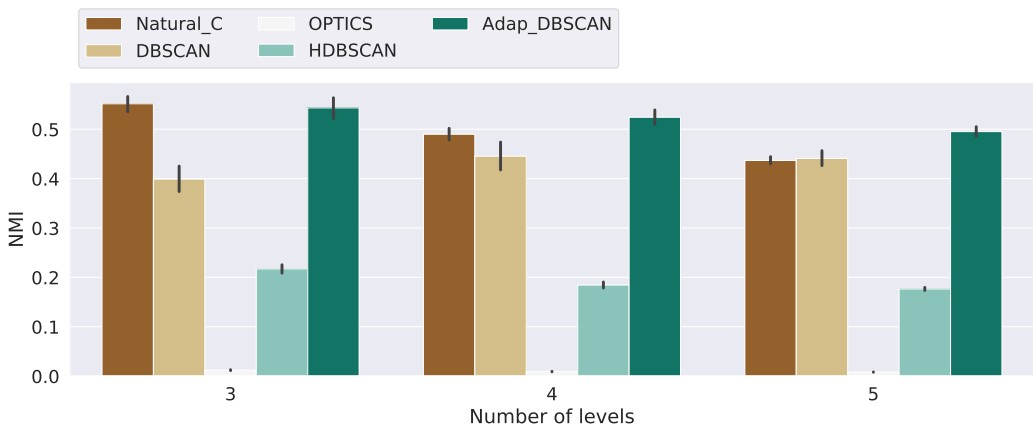

**Figure 8 Normalized mutual information score for the different algorithms.** Results for the normalized mutual information (NMI) for the experiments with 3, 4 and 5 levels. Bars height represent the mean NMI value while the black lines correspond to the 95% confidence interval.

If the polygons have a large Jaccard Similarity and the number of points in the intersections is large, then $SSM$ will also be large. On the contrary, if the polygons have low Jacacard similarity then a penalization will occur and the $SSM$ will have a lower value even if the cardinality of the intersection is large. This means that our measure will penalize clusterizations that over or under separate points in different clusters with respect to the ground truth.

$SSM$ will penalize for points that belong to a $P_l$ for some $l \in \{0, \ldots, n\}$, that are not inside any $Q \in \mathcal{Q}_i^C$. Also a penalization will occur for the points that belong to $Q_k$ for some $k \in \{0, \ldots, m\}$ that not belong to any $P \in \mathcal{P}_i^O$. This is important to ensure that algorithms that properly classify a grater number of points as signal are not penalized.

Thus our $SSM$ allows for the direct comparison between the polygons in the ground truth and those produced by an arbitrary algorithm, producing a global similarity measure for each level that captures not only how similar the polygons are, but also, how many points are in the most similar polygons.

## RESULTS AND DISCUSSION

Results for the $NMI$ metric are shown in Fig. 8, displaying the mean $NMI$ values along with the 95% confidence intervals. These results clearly show the poor performance of HDBSCAN and OPTICS as global classification algorithms, that is, they are not able to distinguish, across hierarchical levels, between noise and signal points. This behavior can be explained by the fact that those algorithms are intended to discover structures that are persistent trough the whole clustering hierarchy, which is different from the problem of finding the nested cluster structure, as explained in 'Clustering Algorithms'. On the other hand, Natural Cities, DBSCAN and Adaptative DBSCAN consistently perform better across the three number of levels tested. In general the performance of Natural Cities and Adaptative DBSCAN is better than DBSCAN and this later exhibits larger confidence

**Table 1** **Pairwise Welch's t-tests for the difference of the experimental means for the NMI metric for all three levels considered.** Values of $t$-test above 2 indicate a significative difference between the observed means.

| Number of Levels | 3 | 4 | 5 |
|---|---|---|---|
| | $t$-test | $t$-test | $t$-test |
| **Algorithms** | | | |
| Natural_C *vs* DBSCAN | −0.46 | 9.37* | 2.89* |
| Natural_C *vs* OPTICS | 108.72* | 68.45* | 79.07* |
| Natural_C *vs* HDBSCAN | 60.97* | 36.48* | 44.71* |
| Natural_C *vs* Adap_DBSCAN | −9.08* | 0.63* | −3.54* |
| DBSCAN *vs* OPTICS | 54.59* | 27.14* | 30.91* |
| DBSCAN *vs* HDBSCAN | 32.65* | 12.12* | 18.06* |
| DBSCAN *vs* Adap_DBSCAN | −5.77* | −8.08* | −4.92* |
| OPTICS *vs* HDBSCAN | −102.13* | −43.45* | −55.77* |
| OPTICS *vs* Adap_DBSCAN | −95.49* | −49.31* | −67.92* |
| HDBSCAN *vs* Adap_DBSCAN | −59.50* | −27.76* | −41.45* |

**Notes.**
An asterisk (*) indicates significative at the 95% confidence level.

intervals, so its results are less consistent across experiments with the same number of levels. It is also interesting to notice that the results are in general consistent across the number of levels tested. In Table 1 we show the results of the pairwise Welch's $t$-test for the differences in the mean NMI values between algorithms. As can be seen, the test indicates that all differences, except that between Natural Cities and Hierarchical DBSCAN are significant.

Results for the balanced accuracy (BA) metric are shown in Fig. 9. Once again, the performance of DBSCAN, Natural Cities and Adaptive DBSCAN is generally better than the performance of OPTICS and HDBSCAN. Adaptive DBSCAN and Natural Cities are the best algorithms at separating signal from noise on a level by level basis, having a similar performance across levels for the three numbers of levels considered. The wider confidence interval of BA values for DBSCAN reflects the rigidity in the way the algorithm selects $\varepsilon$, using the same value for all clusters at the same level, as compared to Adaptive DBSCAN which adapts $\varepsilon$ for each cluster and Natural Cities whose heads-tails break is also computed for each cluster. Table 2 shows the results of the pairwise Welch's $t$-test for the differences in the mean BA values between algorithms. Most of the differences are significative, except for Natural Cities *vs* Adaptive DBSCAN at levels 0, 3 and 4 and OPTICS *vs* HDBSCAN at level 4.

Finally, in Fig. 10 we show the results for the Shape Similarity Measure (SSM). In this case we begin our comparisons in the first hierarchical level, since the level 0 polygons are the same for all algorithms. This results also show the poor performance of OPTICS and HDBSCAN. In this case Adaptive DBSCAN consistently outperforms Natural Cities and DBSCAN, specially as the number of levels increase. The local nature of $\varepsilon$ in Adaptive DBSCAN, coupled with the capacity of DBSCAN to find arbitrarily shaped density based clusters, allows the algorithm to better reproduce the cluster shapes in the ground truth data. Table 3 shows the results of the pairwise Welch's $t$-test for the differences in the mean

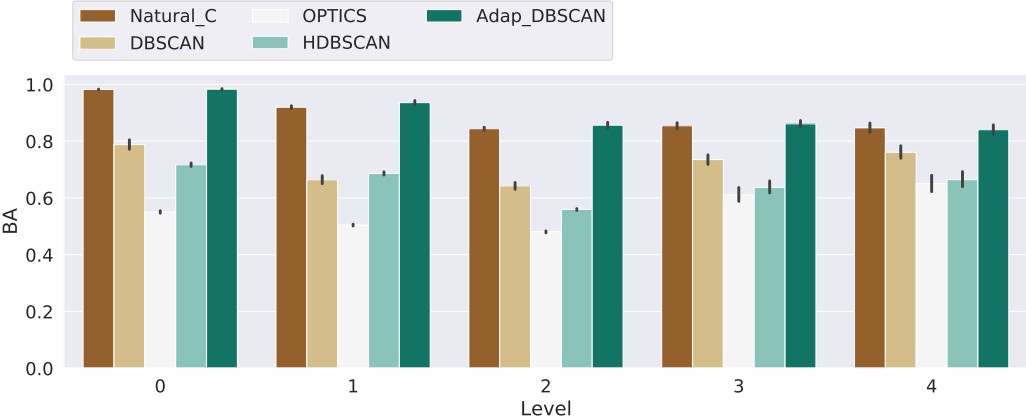

**Figure 9 Balanced accuracy score for the different algorithms.** Results for the balanced accuracy (BA) for the experiments with 3, 4 and 5 levels, the Level axis indicates the BA value for that specific level. Bars height represent the mean BA value while the black lines correspond to the 95% confidence interval.

**Table 2 Pairwise Welch's t-tests for the difference of the experimental means for the BA metric for each level across all experiments.** Values of the $t$-test above 2 indicate a significative difference between the observed means.

| Level | 0 | 1 | 2 | 3 | 4 |
| --- | --- | --- | --- | --- | --- |
| | $t$-test | $t$-test | $t$-test | $t$-test | $t$-test |
| **Algorithms** | | | | | |
| Natural_C *vs* DBSCAN | 23.33* | 32.28* | 32.08* | 11.11* | 6.15* |
| Natural_C *vs* OPTICS | 182.09* | 130.86* | 111.64* | 18.34* | 11.40* |
| Natural_C *vs* HDBSCAN | 79.84* | 59.88* | 88.08* | 17.36* | 10.92* |
| Natural_C *vs* Adap_DBSCAN | −0.53 | −3.49* | −2.05* | −0.82 | 0.54 |
| DBSCAN *vs* OPTICS | 27.48* | 20.89* | 27.42* | 8.23* | 5.88* |
| DBSCAN *vs* HDBSCAN | 7.93* | −2.77* | 14.13* | 6.78* | 5.26* |
| DBSCAN *vs* Adap_DBSCAN | −23.24* | −32.41* | −27.64* | −11.46* | −5.61* |
| OPTICS *vs* HDBSCAN | −41.51* | −54.73* | −32.20* | −1.61* | −0.69 |
| OPTICS *vs* Adap_DBSCAN | −168.32* | −102.33* | −67.58* | −18.55* | −10.92* |
| HDBSCAN *vs* Adap_DBSCAN | −76.72* | −52.18* | −53.55* | −17.59* | −10.43* |

**Notes.**
An asterisk (*) indicates significance at the 95% confidence level.

SSM values between algorithms, again, most of the differences are significant, except for Natural Cities *vs* DBSCAN at level 2, which is clear from the plot on Fig. 10.

## Application on real-world data

As a use case of the proposed technique we will extract the crowd activity patterns for the geolocated Twitter feed for Central Mexico. The test database consists of all geolocated tweets for 2015-01-20, there are 10,366 tweets for this day. Based on the discussion on 'Detection of crowd activity scales and zones' and 'Clustering Algorithms' we will only compare the results obtained with Natural Cities, recursive DBSCAN and Adaptative-DBSCAN. In order to qualitatively compare the results of the algorithms we will focus on

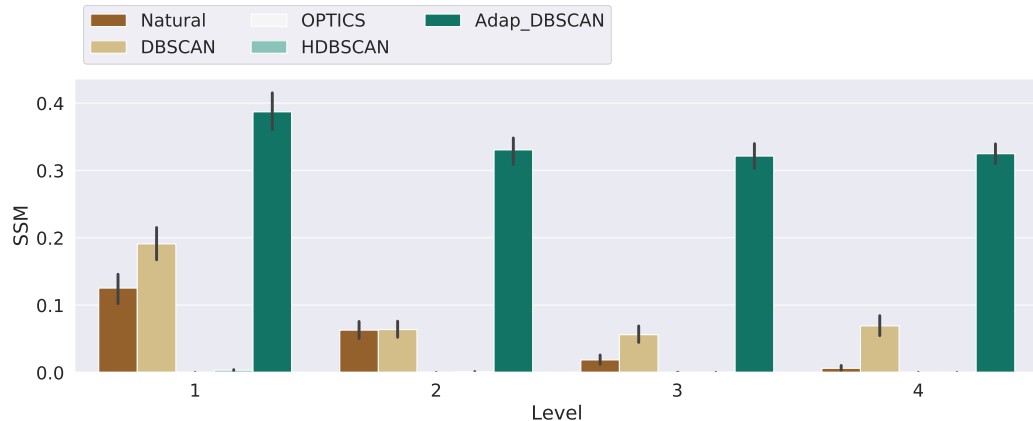

**Figure 10** **Similarity shape measure for the different algorithms.** Results for the similarity shape measure (SSM) for the experiments with 3, 4 and 5 levels, the Level axis indicates the SSM value for that specific level. Bars height represent the mean SSM value while the black lines correspond to the 95% confidence interval.

**Table 3** **Pairwise Welch's t-tests for the difference of the experimental means for the BA metric for each level across all experiments. Values of the $t$-test above 2 indicate a significative difference between the observed means.**

| Level | 1 | 2 | 3 | 4 |
|---|---|---|---|---|
| | $t$-test | $t$-test | $t$-test | $t$-test |
| **Algorithms** | | | | |
| Natural *vs* DBSCAN | −3.82[*] | −0.10 | −5.34[*] | −7.97[*] |
| Natural *vs* OPTICS | 11.00[*] | 9.98[*] | 5.35[*] | 2.87[*] |
| Natural *vs* HDBSCAN | 10.70[*] | 9.85[*] | 5.33[*] | 2.87[*] |
| Natural *vs* Adap_DBSCAN | −14.62[*] | −22.24[*] | −30.38[*] | −40.76[*] |
| DBSCAN *vs* OPTICS | 14.85[*] | 10.45[*] | 9.17[*] | 9.05[*] |
| DBSCAN *vs* HDBSCAN | 14.59[*] | 10.31[*] | 9.16[*] | 9.05[*] |
| DBSCAN *vs* Adap_DBSCAN | −10.39[*] | −22.35[*] | −23.77[*] | −23.84[*] |
| OPTICS *vs* HDBSCAN | −7.72[*] | −3.56[*] | −2.85[*] | 2.21[*] |
| OPTICS *vs* Adap_DBSCAN | −28.00[*] | −32.18[*] | −34.37[*] | −43.09[*] |
| HDBSCAN *vs* Adap_DBSCAN | −27.75[*] | −32.09[*] | −34.36[*] | −43.09[*] |

**Notes.**
An asterisk (*) indicates significance at the 95% confidence level.

two tasks: first the detection of the greatest cities within the region and, second, the detection of the Central Business District (CBD) structure of Mexico City, widely described in the literature (*Escamilla, Cos & Cárdenas, 2016*; *Suarez & Delgado, 2009*). The idea behind this qualitative comparison is to understand how the different algorithms detect known crowd activity patterns at different scales.

Figure 11 through Fig. 13 show the results for the different clustering algorithms. In the larger hierarchical levels, the figures compare our results with the official delimitation of the metropolitan areas, while in the smaller scales, we show the job to housing ratio as an indicator of potential crowd activity. The first thing to notice is that, although the three

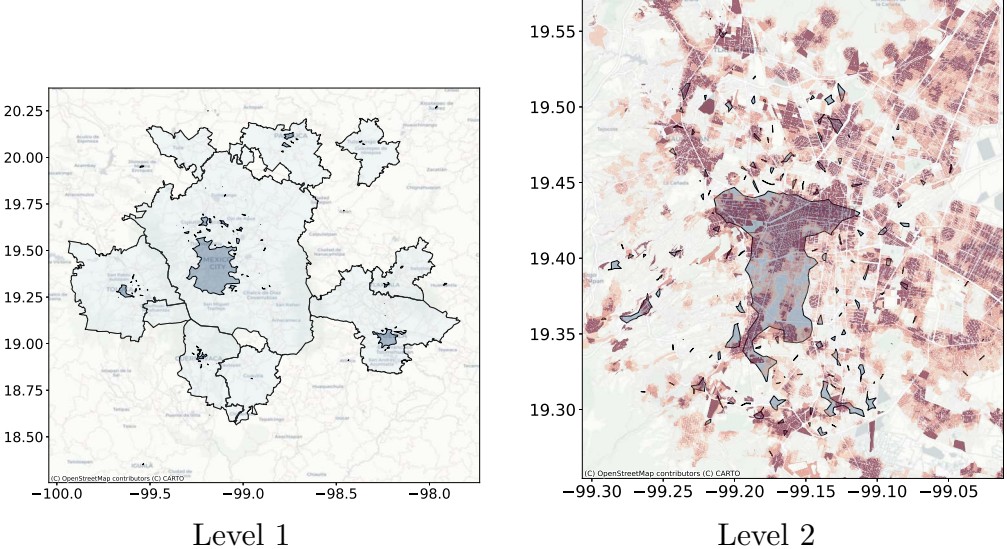

Level 1                    Level 2

**Figure 11    Levels clusters polygons obtained with Natural Cities compared with metropolitan delimitation and compared with job to housing ratio at the city block level.** The left side shows, in a dark color, the polygons obtained with Natural Cities and in clear the official delimitation of metropolitan areas. On the right side, the dark polygons are obtained with Natural Cities and the underlying coloring corresponds to the job to housing ratio at the city block level, darker color indicates a higher job to housing ratio.

algorithms are able to separate the input points into the metropolitan areas, Natural Cities tends to detect the inner cores on the first hierarchical level, while hierarchical DBSCAN and Adaptative-DBSCAN detect larger metropolitan structures. Another interesting finding is that Natural Cities detects the T-shaped pattern of Mexico City's CBD (*Escamilla, Cos & Cárdenas, 2016*; *Suarez & Delgado, 2009*) in two iterations, while hierarchical DBSCAN and Adaptative DBSCAN require 4 and 8 iterations respectively. This means that both DBSCAN based algorithms are detecting intermediate scale structures, while Natural Cities is more *aggressively* reducing the size of the clusters. Another interesting quality in the clusters detected by Natural Cities in contrast to those detected by the other two algorithms is that Natural Cities tends to detect many small clusters at both levels, qualitatively these small clusters do not seem to correspond with any known structure in the city.

For the deeper levels, both in Hierarchical DBSCAN and Adaptative DBSCAN, the patterns detected seem to closely follow the job to housing ratio, this is a good qualitative indicator that the structures detected correspond with the known geographic activity patterns. To further stress this point, in Fig. 14 we show the different clusters detected with the three algorithms for afternoons (14:00 to 18:00) and evenings (18:00 to 22:00), all figures exhibit more concentrated patterns for the evenings, reflecting people gathered at their workplaces. It is also interesting that all algorithms seem to follow the T-shaped pattern of the CBD and also detect some smaller job centers to the South and West, but Natural Cities detects many smaller clusters that do not seem to correspond with relevant geographic structures.

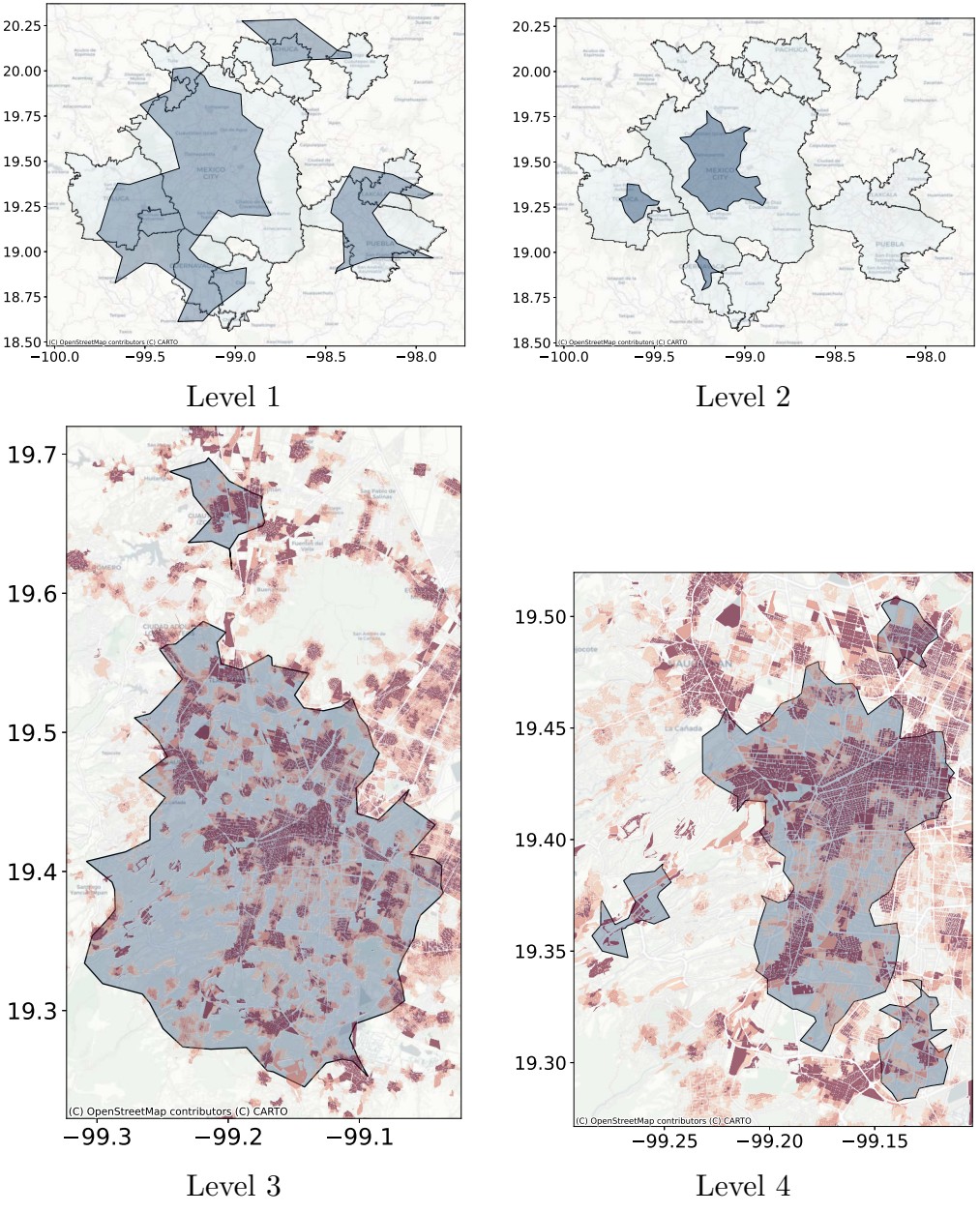

**Figure 12 Levels cluster polygons obtained with DBSCAN compared with metropolitan delimitation and compared with job to housing ratio at the city block level.** Cluster polygons obtained with DBSCAN compared with the metropolitan delimitation and job to housing ratio at the city block level. (A) The dark polygons are the clusters obtained with DBSCAN and the gray polygons are the official delimitation of metropolitan areas. (B) The dark polygons are the clusters obtained with DBSCAN at Level 2 and the gray polygons are the official delimitation of metropolitan areas. (C) The blue polygons are obtained with DB-SCAN at Level 3 and the coloring corresponds to the job to housing ratio at the city block level. Red represents higher jobs to housing ratio. (D) The blue polygons are obtained with DBSCAN at Level 4 and the underlying coloring corresponds to the job to housing ratio at the city block level. Red represents higher jobs to housing ratio.

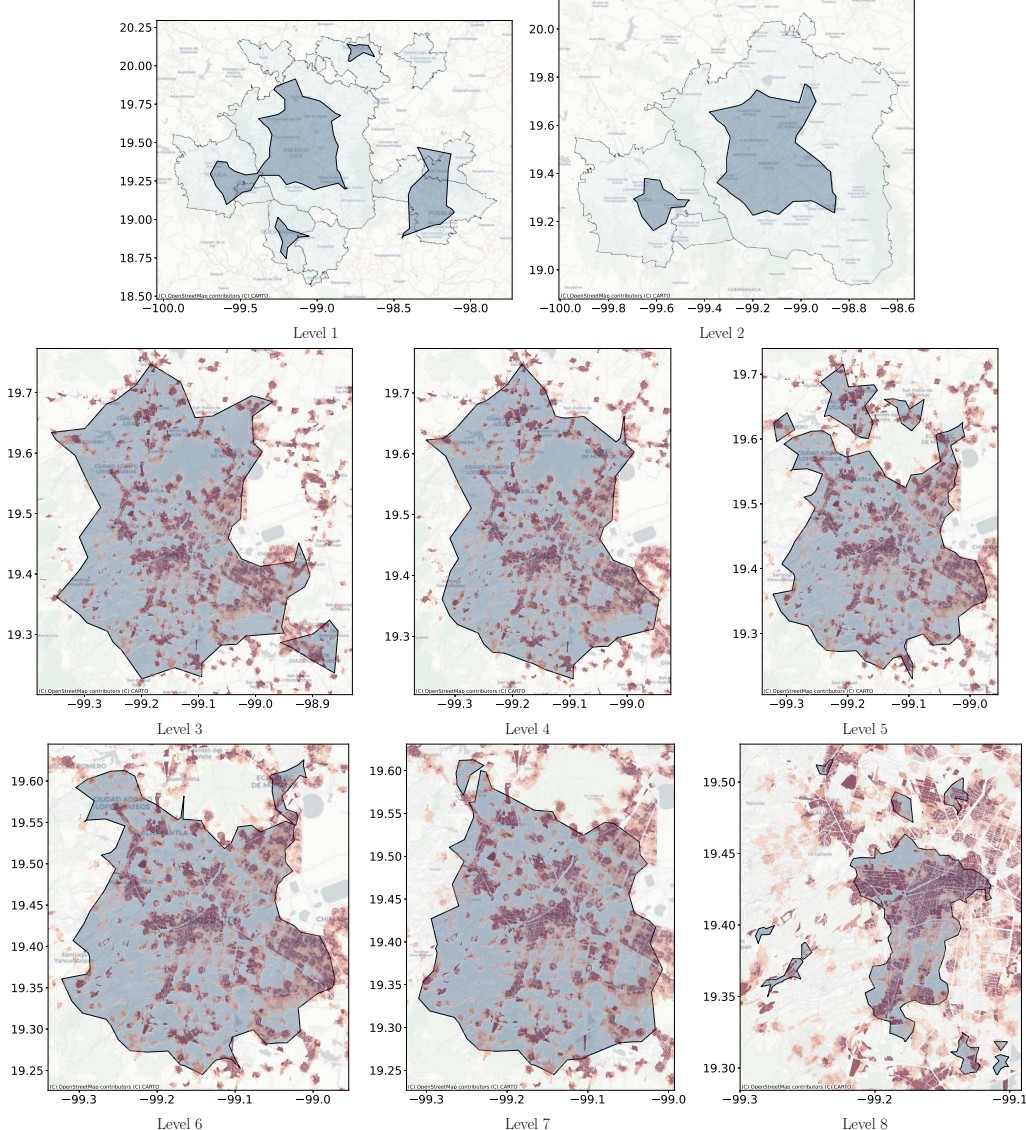

**Figure 13 Level clusters polygons obtained with Adaptative DBSCAN compared with metropolitan delimitation and at lower scale compared with job to housing ratio at the city block level.** (A) The blue polygons are the clusters obtained with Adaptative DBSCAN and the grey polygons are the official delimitation of metropolitan areas. (B) The blue polygons are the clusters obtained with Adaptative DBSCAN and the grey polygons are the official delimitation of metropolitan areas. (C) The blue polygons are obtained with Adaptative DBSCAN and the coloring corresponds to the job to housing ratio at the city block level. Red represents higher jobs to housing ratio. (D) The blue polygons are obtained with Adaptative DBSCAN and the coloring corresponds to the job to housing ratio at the city block level. Red represents higher jobs to housing ratio. (E) The blue polygons are obtained with Adaptative DBSCAN and the coloring corresponds to the job to housing ratio at the city block level. Red represents higher jobs to housing ratio. (F) The blue polygons are obtained with Adaptative DBSCAN and the coloring corresponds to the job to housing ratio at the city block level. Red represents higher jobs to housing ratio. (G) The blue polygons are obtained with Adaptative DBSCAN and the coloring corresponds to the job to housing ratio at the city block level. Red represents higher jobs to housing ratio. (H) The blue polygons are obtained with Adaptative DBSCAN and the coloring corresponds to the job to housing ratio at the city block level. Red represents higher jobs to housing ratio.

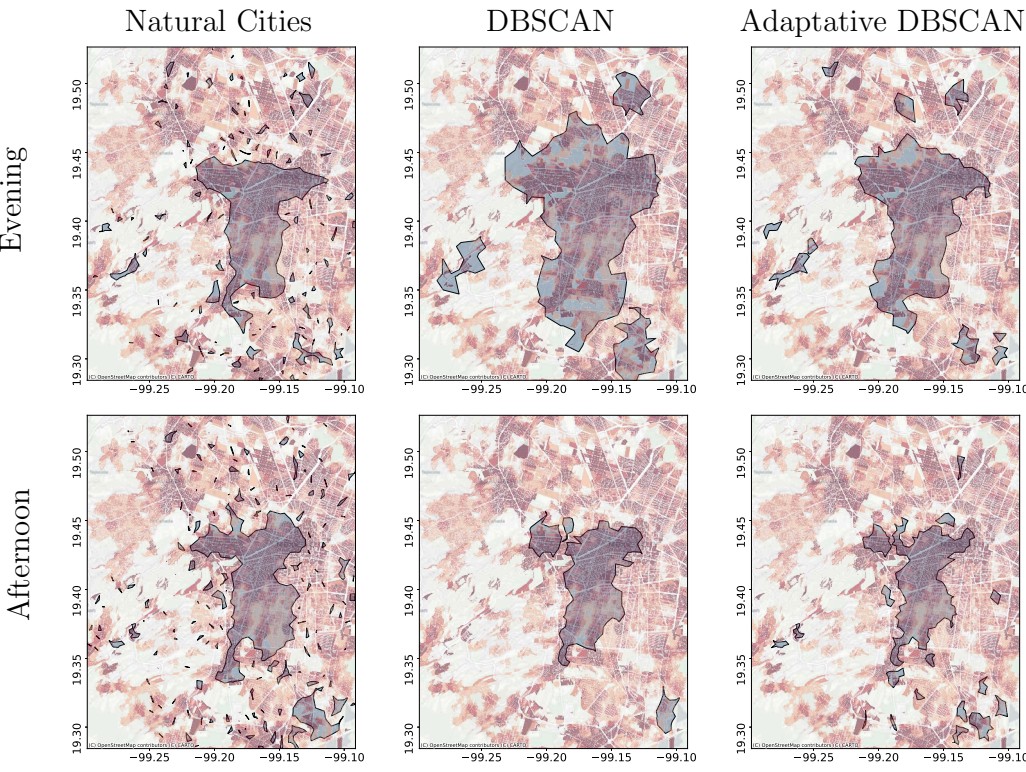

**Figure 14** **Clusters obtained using Natural Cities, DBSCAN, and Adaptative DBSCAN on data for the evening and afternoon.** (A) The clusters obtained using Natural Cities with data for the evening (18:00 to 22:00). (B) The clusters obtained using DBSCAN with data for the evening (18:00 to 22:00). (C) The clusters obtained using Adaptative DBSCAN with data for the evening (18:00 to 22:00). (D) The clusters obtained using Natural Cities with data for the afternoon (14:00 to 18:00). (E) The clusters obtained using DBSCAN with data for the afternoon (14:00 to 18:00). (F) The clusters obtained using Adaptative DB-SCAN with data for the afternoon (14:00 to 18:00). The figures focus on the levels showing the Central Business District area and display in red tones the job to housing ratio.

# CONCLUSIONS AND FURTHER WORK

In this article we presented a synthetic data generator that reproduces structures commonly found on geographical events data sets; introduced a new method, based on the recursive application of DBSCAN coupled with an adaptative algorithm for selecting appropriate values for $\varepsilon$ for each cluster, for detecting hierarchical structures within structures; and presented a general evaluation framework for the comparison of hierarchical crowd activity structures detection algorithms. The results presented in 'Results and discussion' show the poor performance of OPTICS and HDBSCAN for iteratively finding structures within structures in hierarchically structured data. This result shows that the task of finding a hierarchical structure on geographical data is different from the task of finding the most significant cluster structures, confirming the need to develop and evaluate methods specifically tailored to find such structures. The performance of Natural Cities and Adaptative DBSCAN when compared to that of DBSCAN show the importance of considering the local properties of data.

Testing the algorithms against synthetic data allows us to have a fair and objective comparison instead of relying on qualitative and probably subjective evaluations, we believe this is an important step in the further development of tools to uncover hierarchical structures on geographical data. However, in 'Application on real-world data' we tested Natural Cities, Adaptative and Hierarchical DBSCAN against a real world dataset of geolocated Twitter messages, showing that the three algorithms are able to detect the known activity patterns in Mexico City.

In the future we will use the Adaptative DBSCAN algorithm presented in 'Adaptative DBSCAN' to process data such as 911 geolocated calls (police reports) and 311 reports (public service requests) to develop unusual activity detection algorithms, able to capture unusual activity at different scales. Another interesting avenue for further research is the incremental training of the algorithms, finding structures incrementally as more data is fed into the algorithms. Finally, it is also necessary to explicitly incorporate the time dimension and develop algorithms to uncover hierarchical spatio-temporal structures.

### Funding
The authors received no funding for this work.

### Competing Interests
The authors declare there are no competing interests.

### Author Contributions
- J. Miguel Salazar conceived and designed the experiments, performed the experiments, analyzed the data, performed the computation work, prepared figures and/or tables, authored or reviewed drafts of the paper, and approved the final draft.
- Pablo López-Ramírez conceived and designed the experiments, performed the experiments, analyzed the data, performed the computation work, prepared figures and/or tables, authored or reviewed drafts of the paper, and approved the final draft.
- Oscar S. Siordia conceived and designed the experiments, analyzed the data, authored or reviewed drafts of the paper, and approved the final draft.

### Data Availability
The libraries are available at GitHub: https://github.com/CentroGeo/HierarchicalGeoClustering

The experimental setup is available at GitHub: https://github.com/msalazarcgeo/HGC_Article.

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
