# Peer review of "Detection of hierarchical crowd activity structures in geographic point data"

_PeerJ Computer Science, doi:10.7717/peerj-cs.978_

## Round 0.1 · original submission · Major Revisions

The reviewers have identified some merits of the paper. But they also pointed out some shortcomings. A revision is needed. Please provide a detailed response letter together with your revision. Thanks.

Reviewer 1 ·

Basic reporting

This paper presents a synthetic data generator that reproduces structures commonly found on geographical event data sets and describes clustering and hierarchical scale extraction algorithms to be evaluated. Basically, the topic is interesting and the manuscript is well-written but I have some comments to be improved.

Major comments:
- The authors mentioned that the second contribution is to propose an improvement of the DBSCAN algorithm, but existing clustering algorithms were introduced and used. Please clarify improvement points of the algorithm. The title should also be modified to fit the content because the current one seems to be the proposal of an algorithm.

Minor comments:
- Figure 1 should be explained more detail such as the meaning of colors. Also, the quality of the Figure 1(b) should be improved.
- The authors discussed the results in Section 7, so the section name should be Results and discussions.
- There are some typos as follows.
Line 20: “even data” would be “event data”
Line 166: “as shown in 1” would be “as shown in Figure 1”
Line 431: There is a space missing after the comma.
Line 492: “This results” should be “These results”

Experimental design

no comment

Validity of the findings

no comment

Reviewer 2 ·

Basic reporting

1. There are several places (Lines 131-137, Lines 186-191, Lines 494-496) where it is argued that the expected differences between different clustering algorithms (e.g., HDBSCAN and OPTICS vs. DBSCAN and its adaptive version). It would be beneficial to use a toy example dataset to demonstrate the differences argued here.
2. Some figures (e.g., Figure 1) need to be of higher resolution to be legible (check all other figures for this issue). In Figure 1 caption, please describe how (a) corresponds with (b). For example, is a node in (b) a point in (a)?
3. Lines 160-161: The reviewer did not quite understand what “three iterations of head-tail breaks” entails. Please elaborate.
4. Line 249: What important factors need to be considered when manually setting parameters?
5. Line 518: Does this "new method" refer to applying clustering algorithms recursively to discover structure within structure in general, or specifically to applying the adaptive DBSCAN recursively? Need to be clear.

Experimental design

1. Line 404: “statistically significant evaluations”. What does this entail exactly? Later in the results section, only box plots are used to show the differing performance of the algorithms. I would expect some sorts of statistical significance test on the differences (or comparison of confidence intervals?).
2. Lines 530-531: The reviewer thinks it is necessary to do so in this paper. Apply the adaptive DBSCAN a real-world dataset recursively to discover structures within structures, and see if the results make sense (probably with only qualitative evaluations).

Validity of the findings

1. Line 404: “statistically significant evaluations”. What does this entail exactly? Later in the results section, only box plots are used to show the differing performance of the algorithms. I would expect some sorts of statistical significance test on the differences (or comparison of confidence intervals?).
2. Lines 530-531: The reviewer thinks it is necessary to do so in this paper. Apply the adaptive DBSCAN a real-world dataset recursively to discover structures within structures, and see if the results make sense (probably with only qualitative evaluations).

Additional comments

See the annotated pdf file attached for other minor comments.

Annotated reviews are not available for download in order to protect the identity of reviewers who chose to remain anonymous.

---

## Round 0.2 · accepted · Accept

The paper can be accepted. Congratulations.

Reviewer 1 ·

Basic reporting

no comment

Experimental design

no comment

Validity of the findings

no comment

Additional comments

Thank you for your revisions. I checked the revised manuscript and confirmed that the authors have addressed my major and minor comments. I think the paper is accepted for publication after minor revision below.

1. Figure 1: Instead of arranging figures vertically, how about arranging them horizontally?
2. The caption of Figure 14: hours should be removed.

Reviewer 2 ·

Basic reporting

The revised manuscript has sufficiently addressed my comments. I now recommend acceptance for publication.

Experimental design

The revised manuscript has sufficiently addressed my comments. I now recommend acceptance for publication.

Validity of the findings

The revised manuscript has sufficiently addressed my comments. I now recommend acceptance for publication.

Additional comments

The revised manuscript has sufficiently addressed my comments. I now recommend acceptance for publication.